# An EMG-to-Force Processing Approach to Estimating Knee Muscle Forces during Adult, Self-Selected Speed Gait

**DOI:** 10.3390/bioengineering10080980

**Published:** 2023-08-20

**Authors:** Ross Bogey

**Affiliations:** Department of Physical Medicine and Rehabilitation, Western University of the Health Sciences, 309 East 2nd Street, Pomona, CA 91766, USA; rosssbogey@gmail.com

**Keywords:** model, muscle force, concentric contraction, eccentric contraction, normal gait

## Abstract

Background: The purpose of this study was to determine the force production during self-selected speed normal gait by muscle–tendon units that cross the knee. The force of a single knee muscle is not directly measurable without invasive methods, yet invasive techniques are not appropriate for clinical use. Thus, an EMG-to-force processing (EFP) model was developed which scaled muscle–tendon unit (MTU) force output to gait EMG. Methods: An EMG-to-force processing (EFP) model was developed which scaled muscle–tendon unit (MTU) force output to gait EMG. Active muscle force power was defined as the product of MTU forces (derived from EFP) and that muscle’s contraction velocity. Net knee EFP moment was determined by summing individual active knee muscle moments. Net knee moments were also calculated for these study participants via inverse dynamics (kinetics plus kinematics, KIN). The inverse dynamics technique used are well accepted and the KIN net moment was used to validate or reject this model. Closeness of fit of the moment power curves for the two methods (during active muscle forces) was used to validate the model. Results: The correlation between the EFP and KIN methods was sufficiently close, suggesting validation of the model’s ability to provide reasonable estimates of knee muscle forces. Conclusions: The EMG-to-force processing approach provides reasonable estimates of active individual knee muscle forces in self-selected speed walking in neurologically intact adults.

## 1. Introduction

Normal walking consists of the sequential graded contraction and relaxation of the bilateral lower extremity muscles. Our understanding of normal gait would be enriched if the forces produced by individual muscles forces during self-selected speed walking were known. This information would give clinicians and scientists the tools to have a more complete grasp of the underlying neural control of movement. Unfortunately, determining in vivo muscle forces is complicated.

Most lower extremity joints are actuated by more muscles than necessary to perform basic movement(s). However, the rationale for this surfeit of available muscles crossing each joint is not obvious. As a result of this mathematical indeterminacy, the equations of motion alone are not sufficient to determine the force in each muscle during movement. Thus the leg muscles are typically assessed as synergistic groups for analysis purposes, and the joint moment is presumed to be correlated to the magnitude of muscle forces that span the joint [1]. One difficulty with this approach is that muscles within a functional group may not share timing or role(s) during walking. An example is the quadriceps functional group. Of the four muscles comprising this group, only the rectus femoris crosses the hip, and its role may vary dramatically from the that of the three vastus muscles which extend the knee during walking [2]. Other methods that address individual muscle forces are certainly needed.

The assessment of muscle forces and the role of individual muscles is sometimes resolved by resorting to some form of optimization criteria. Here a single cost function—such as metabolic cost, muscle stress or muscle endurance—is typically minimized or maximized as kinematic data are combined with specified objective function assumptions [3,4,5,6,7,8]. While this approach has increased our understanding of gait mechanics, an unresolvable drawback is that the determination of objective functions remains subjective and there is no direct means of experimental validation [9].

An EMG-to-force processing approach has been successfully used to estimate ankle [10,11,12,13,14] and hip [15] muscle forces in normal gait. Hof and colleagues [10,11,12,13] published a series of insightful studies that gave realistic estimates of plantar flexor forces across several tasks, including walking. Bogey and colleagues [14] augmented this approach in determining the gait muscle forces for all ankle muscles examined by Hof [10,11,12,13] plus the remainder of the ankle muscles. Their plantar flexor force estimates were similar to those obtained by Hof when examining level walking [16]. Further, the plantar flexor force estimates of both Hof [10,11,12,13] and Bogey [14] were comparable to level, self-selected speed gait muscle–tendon unit forces obtained directly from a force transducer attached to the Achilles tendon [17,18]. Hip muscle forces have also been determined under these same walking conditions [15]. Yet, direct measurement of in vivo hip or knee muscle forces was not possible due to mechanical constraints. To date, no one has directly measured adult human patellar tendon forces or individual hamstring muscle forces during normal walking. Substantial knee muscle forces are produced during a limited portion of the gait cycle, and the forces produced by any of these muscles have not been successfully described previously.

One goal of this study was to offer insight into knee muscle forces (and actions) during gait. Modeling and muscle force simulations have the potential to improve patient care for movement-related disorders. Reliable estimations of these in vivo forces may enhance our understanding of the muscle force requirements for efficient adult walking and help optimize future rehabilitation protocols aimed at the recovery of near-normal walking after injury or disease.

Knee muscle force values during self-selected speed walking were determined by (i) inverse dynamics and (ii) an EMG-to-force processing (EFP) model during the same gait trial for each study participant. The net knee moment is the sum of the individual moments about that joint from the forces developed by muscles (and possibly other structures crossing the knee [19]). The net knee moment for each method was statistically compared at one percent gait cycle intervals. Closeness of fit between the EFP and inverse dynamics power curves were used to validate the EFP model. If the resulting moment curves were sufficiently similar, the EFP moment estimates were deconstructed to analyze the gait forces produced by each muscle–tendon unit crossing the knee.

## 2. Subjects

A convenience sample of 18 adult males with no history of neuromusculoskeletal disease were recruited to participate in the study. The study participants had a mean age of 27 ± 3.2 years (range 23–34 years), and a mean mass of 73.9 kg (±6.6 kg). Due to the extensive number of muscles examined, each study participant performed two self-selected speed walking trials, with data selected from each trial. The time between first and second gait analyses did not exceed 15 days for any study participant. In normal adults, the gait parameters examined are highly consistent across trials [20], and the need to combine data from multiple trials did not confound the results. The mean walking speed for these study participants was 82.0 ± 3.6 m per minute. Subjects consented to participate following explanation of the procedure and review of the informed consent, as approved by the Institute Review Board, and signed the Rights of Human Subjects form (Figure 1).

## 3. Methods

### 3.1. Kinematics Plus Kinetics (KIN)

Gait data acquisition and processing steps used in this study have been described in greater detail elsewhere [14]. The primary focus was the knee moment for the entire walking cycle. Stance and swing times, contact forces and moments (kinetics), limb-segment kinematics and EMG data were simultaneously acquired while study participants performed self-selected speed level walking (Figure 1). Normal gait consists of two phases: stance phase and swing phase. These phases can be further divided into a total of eight sub-phases [21]. The stance phase normally occupies about 60% of the total gait cycle [21] during which at least some part of the ipsilateral foot is in contact with the ground. The swing phase occupies the remaining 40% of the gait cycle [21]. Beginning with quiet standing, each study participant reached their comfortable (self-selected) walking speed after a few strides. They maintained this pace across the middle six meters of the test area then decelerated prior to reaching the end of the 12-meter walkway. Gait analysis was limited to the middle six meters of the trial where walking velocity was essentially constant. Study participants wore appropriate clothing, which consisted of a light shirt, shorts and their own flat-soled shoes. Contact-closing footswitches [21] were placed in the participant’s shoes during the trials to determine stance and swing times. Round reflective markers were placed over pre-determined anatomic landmarks to determine joint centers [22].

Study participants performed several preliminary walking trails to become acclimated to the test environment. Since normal gait is essentially symmetric [23], only data from the right leg were analyzed. Data were averaged across five noise-free trials for each study participant.

Motion data were sampled at 100 Hz with an eight-camera system (Motion Analysis Corporation Model Hawk). Marker coordinates were bi-directionally smoothed using a fourth-order Butterworth filter. The effective cutoff frequency was 6 Hz. Joint angles, joint velocity and acceleration, relative limb segment positions and limb segment linear velocities and accelerations were determined as described elsewhere [24]. Paired force platforms (AMTI type OR6–7) were used to collect ground reaction force (GRF) data. GRF data were acquired at 600 Hz and smoothed. The center of pressure was determined using well-accepted methods described elsewhere [24]. Ground reaction force and motion data were temporally matched. Inertial properties of the limb segments were based on the methods first described by Dempster [25]. Knee moments were computed via application of Newtonian mechanics using in-house software that was independent of the motion capture or force platform system used.

### 3.2. EMG-to-Force Processing—Force Processor

The neuromusculoskeletal model included lower extremity skeletal structures plus 14 muscles crossing the knee. The force-generating properties, path geometry and MTU attachment sites were based on data reported elsewhere [26,27]. The model consisted of three-space representations of the bones and muscle–tendon paths, kinematic descriptions of the ankle, knee and hip, plus a nominal biomechanical model of each musculotendinous unit (MTU).

Each muscle–tendon unit was scaled to estimate the isometric force-generating properties of the MTU, based on a Hill-type muscle in series with an elastic tendon [28] (Figure 2). For nearly a century, the classical Hill model has functioned as an excitation–contraction coupling model with three distinct elements—one contractile element (activated by neurological input (EMG)) and two non-linear springs. One of the spring elements is in series (Series Elastic Component, representing the inherent elasticity of the myofilaments) and one is in parallel (Passive Elastic Component, representing the mechanical behavior of the non-contractile tissues). The presence of the three components categorizes the total stress into a passive part and an active part.

The effects of muscle force–velocity relations, muscle contraction history and contraction type were considered [29,30,31].

### 3.3. METHODS: EMG-to-Force Processing—Dynamic Electromyography (EMG)

Electromyographic (EMG) activity was obtained from 14 muscles crossing the knee—vastus lateralis (VL), vastus intermedius (VI), vastus medialis oblique (VMO), rectus femoris (RF), semitendinosis (STEND), semimembranosis (SMEMB), long head of the biceps femoris (BFLH), short head of the biceps femoris (BFSH), sartorius (SART), gracilis (GRAC), tensor fascia lata (TFL), medial gastrocnemius (MGAST), lateral gastrocnemius (LGAST) and popliteus (POP). Electromyographic activity was recorded with insulted bipolar 50µ stainless steel wire electrodes. The insulation was removed 2 mm from the wire ends, and the tips curved slightly to limit migration of the wires within the muscle. Electrodes were inserted into each knee muscle with a 20-gauge hypodermic needle [32]. Electrodes were inserted near the presumed motor point of the muscle [33]. A ground electrode was placed over a convenient lower extremity bony landmark. Electrode placement was confirmed by voluntary muscle contraction and electrical stimulation of the muscle via the indwelling electrode.

The EMG system bandwidth was 10–1000 Hz, with an overall gain of 1000 [34]. EMG was normalized to a maximum voluntary contraction (MVC [35]). The limbs were positioned for each muscle test as described by Hislop [36]. A board-certified rehabilitation physician (RB) conducted all the manual muscle testing. The manual muscle test was five-seconds in duration, and the one-second interval with the highest mean voltage was established as the one hundred percent maximum voluntary contraction reference value (100%MVC). A second maximum muscle test was performed at the end of each test to assure continued integrity of the electrode insertion.

The gait cycle interval and foot support patterns were recorded with footswitches. Both EMG and footswitch data were collected at 2500 Hz. The digitized EMG data were rectified. Baseline noise was corrected with a resting run. Normalization of stride data were accomplished by determining the %MVC value for each 1%GC interval.

EMG input to the force processor model required two sequential processing steps. The initial step was determining each muscle’s EMG linear envelope on a stride-by-stride basis [23]. For this, the raw EMG signal for each stride was rectified, and the mean relative intensity was determined for the first 50-sample interval. A new 50-sample mean was established by advancing the analysis window by one data point. This moving-window averaging process was repeated for the entire gait trial. Using a linear interpolation method, the several thousand EMG data points per stride were reduced to one hundred EMG relative intensity values—one for each percent gait cycle. The within-subject EMG profiles were then determined from individual stride EMG data using the methodology described by Bogey [37].

There is a timing mismatch between a muscle’s electrical activity and force output. The lag between electrical activity and force production (electromechanical delay, EMD) may be influenced by propagation of the action potential on the muscle membrane, excitation–contraction coupling events and elongation of the series elastic component by the contractile element [38]. EMG onset activity leads muscle force production by slightly less than 20 ms [39]. Muscle activation (EMG) and force production continue to be temporally offset during the contraction, then muscle force is maintained well after that muscle’s EMG activity has returned to baseline. The “relaxation EMD” may last as long as 250–275 ms [39], thus the selection of a constant offset correction between EMG timing and predictions of force generation (e.g., 50 ms) would induce error into a neuromuscular model. Zajac [40] modeled EMD using a second-order partial differential equation model. In contrast, Bogey [14] used a second order critically damped Butterworth low-pass digital filter to temporally match EMG activity and muscle force. Both methods lead to similar activation dynamics, and the latter method was used here. The selected approach results in a relatively short onset EMD and more prolonged relaxation EMD. The EMG input to the model (activation dynamics, ((ε(t)) was defined by
ε(t) = γ[κ_1_E(t) + κ_2_E(t − 1) + κ_3_E(t − 2) + κ_4_ε(t − 1) + κ_5_ε(t − 2)].(1)

In Equation (1), *E* represents the percent maximum voluntary contraction for the within-subject EMG profiles, and *e* represents the percent maximum voluntary contraction for the within-subject EMG profiles following filtering. The use of the constant *γ* (=*E_MAX_*/*e_MAX_*) minimized further attenuation of the EMG signal due to signal processing. The values of the constants *κ*_1_–*κ*_5_ were derived from the sampling frequency-to-cutoff frequency ratio (F_s_/F_c_), and a single F_s_/F_c_ ratio was empirically determined [41,42].

Muscle–tendon unit length (*L^MTU^*) and force (*F^MTU^*) were determined at each percent gait cycle interval. Muscle shortening or lengthening contraction velocity was established by mathematical differentiation of MTU length (*v^MTU^ = dL^MTU^*/*dt*). Positive velocities were related to concentric (shortening) muscle contractions.

The EFP net knee moment for each study participant was obtained by summing the moment produced by each muscle crossing the knee. Summed values for all 14 muscles were determined at 1%GC intervals. The correlation between the across-subject mean EFP knee moment and the mean kinetics plus kinematics knee moment for these same study participants was performed, where
(2)$$M_{knee}^{KIN}\approx M_{knee}^{EFP}$$

### 3.4. Statistics

#### 3.4.1. Gait Duration

Review of inverse dynamics knee moments (KIN) demonstrated that about one-third of the gait cycle had non-zero knee moments yet absent or low relative amplitude EMG activity (Figure 2). Knee motion has been shown to occur due to forces and moments at adjacent joints [43,44,45]. The vigorous plantar flexion observed in pre-swing leads to substantial knee flexion, as part of a closed kinetic chain [46]. Next, early swing is notable for the activation of ipsilateral hip flexors and associated knee flexion [15]. As a result, pre-swing and early swing knee flexion is essentially passive (no muscle forces required). As the swing phase continues, hip flexion slows, and knee extension is generated due to transfer of energy between linked leg segments [47]. As the goal of this analysis was to obtain reasonable estimates of the knee moments due to only knee muscle forces, the analysis was limited to those periods where knee muscle EMG (and force) was shown to be present. Accordingly, the assessment period began in the early swing phase (≈68%GC) and concluded when all quadricep and/or hamstring activity ended (≈32% of the subsequent stride) (Figure 3). Thus, it follows that each repeating, EMG-activity period was about 770 ms in duration, and approximately two-thirds (64%GC) of the gait duration in each stride had substantial knee force production.

EMG activity was recorded from several muscles that cross the knee and affect the net knee moment, yet their force production has been reported elsewhere [2,15]. The rectus femoris (RF) EMG was out of phase with the other knee extensors that insert into the patellar tendon (VL, VI, VMO). RF is unique among the knee extensors as it is biarticular. Previous work has demonstrated its role in gait as a hip flexor much more than a knee extensor [15]. Many leg muscles have what might be described as “sub-compartments”; that is, aspects of these muscles may have unique dual innervation (e.g., adductor magnus [48]) or compartmentation of muscle fibers with possible unique function for each muscle segment (e.g., gluteus medius [49]). The vastus lateralis (VL) has a divided neural input possibly capable of separately innervating the upper lateral portion and medial lower portion of the muscle [50]. However, it is unclear whether these fibers can or do function independently in gait, and in this examination, the segments were represented as a single muscle.

Tensor fascia lata can produce a modest amount of knee extensor force and is included in this analysis to compare EFP and KIN methods, only. TFL force production has been reported elsewhere [51]. Finally, MGAST and LGAST produce force in pre-swing that directly leads to knee flexion. Neither produces a force in swing, and thus do not directly contribute to the knee flexion observed during that gait period. The force [14] and power [52] output by these muscles has been reported elsewhere.

#### 3.4.2. Assessment of Model (Analysis)

The Pearson correlation coefficient (r) was used to measure the variance of the two values at the knee, as inspection of pilot data indicated that the model and experimental data satisfied parametric statistical assumptions [53]. An r ≥ 0.80 was required for EFP model acceptance. This number, consistent with other modeling approaches [14], represents a high level of equivalence between the model and experimental values.

## 4. Results

EFP and KIN knee moments were closely related, which satisfies Equation (2) (r = 0.91, Figure 3). The peak difference in knee moments was 32 Nm and occurred at the transition from swing to stance (99%GC). The mean difference (“error”) was 10.9 Nm.

Figure 4, Figure 5 and Figure 6 show the force output by all knee muscles. Note that the hamstring muscles (BFLH, SMEMB, STEND) were essentially in-phase (Figure 4), as were the vasti muscles (VL, VI, VMO) (Figure 5), suggesting a synergistic role during gait for each group. Other knee muscle forces were more variable and do not appear to endorse synergistic activity with either the hamstrings or primary knee extensors (Figure 6). For most (≈68% of study participants), POP is active throughout the gait cycle. Several unique EMG activity profiles were demonstrated by the remaining study participants for this muscle. Popliteus is not part of a synergistic group and can produce only nominal amounts of force at any point in the gait cycle (Figure 7). Peak force production for POP was less than 3% of the maximum force by either SMEMB or VL.

## 5. Discussion

### 5.1. Closeness of Fit

Knee muscle forces are absent during much of the walking cycle. The knee is commonly modeled as a passive multi-link pendulum [16]. The body’s center of mass travels forward over the fixed foot during stance, and knee extension in late-single support is largely passive [46]. The knee flexion that temporally follows in pre-swing is also passive—that is, no knee flexor forces are required. Next, swing limb hip flexion induces early swing knee flexion. Later in swing, the knee begins extension, prior to the onset of knee extensor muscle forces. One theory is that this passive knee extension is the result of gravitational forces [46], yet gravity alone cannot account for the observed knee motion as gravitational forces would accelerate all swing phase limb components downward. Early swing phase motion can be influenced by muscles that cross the knee plus movements at other joints [43,45]. Hence, both velocity-dependent and muscular forces (generated at other joints) contribute to knee motion in gait. Due to the aforementioned factors, a substantial portion of the gait cycle does not require forces by muscles that in fact cross the knee. For this reason, the comparison of the EFP method to accepted inverse dynamics techniques was limited to the gait epoch where considerable muscle forces—and likely only muscle forces—lead to knee motion. During that interval, the closeness of fit between the EFP and KIN curves indicates that the muscle-force estimates are reasonable. One caveat, however, is the assumption that the inverse dynamics of the knee moments are precise. Assumptions about segment masses and radii of gyration have built-in error when literature-obtained values are used to estimate segment masses and the distribution of that mass in the individual limb segments. However, direct measurement of knee muscle forces is not possible with current technology. Thus, confirmation of muscle forces must be inferred from the joint moment.

### 5.2. Force Production by Individual Muscles

Hamstring force production began in the second half of swing, as hip flexion slows, and knee extension begins (Figure 4). STEND and SMEMB onset is nearly identical, with SMEMB peak force >> STEND. The BFLH onset is slightly delayed (about 80 ms), with cessation of all three muscles in the next stride’s single support phase (≈30%GC). These bi-articular muscles act to slow the swing phase knee extension, then transition to extending the hip in the subsequent weight-acceptance phase [2]. Similar findings were previously presented by Frigo and colleagues [54]. Using a dynamic model of knee flexors and extensors, they noted that the co-contraction of knee extensors and hamstrings (seen in stance phase) enhanced the hip extensor effect of the hamstrings. All hamstring forces have ended by mid-single support.

Time of force onset, peak force and force cessation are nearly identical for the vastus muscles. VL and VMO peak force are each 100 N greater than VI. All begin activity in late swing. Late-swing phase knee extensor activity is likely needed to maximize the step length, and to prepare these muscles for their role in attenuating knee flexion in the subsequent weight acceptance phase of the next stride. Their force production continued through weight acceptance. Knee extensor forces systematically decreased in single support as the knee was extending (15–40%GC, Figure 5). The combination of decreasing knee extensor muscle forces with knee extension suggests that stance phase knee extension is principally passive. The center of mass is moving in a posterior-to-anterior direction over the fixed foot during this phase, and knee extension is required to keep the integrity of the connected lower limb segments. Thus, no force by the knee extensors is required to extend the knee in mid-stance. It is notable that late-single support, pre-swing and early swing (≈500 ms duration) have no force output by any hamstring or vastus muscles.

SART, TFL, GRAC and BFSH produce markedly lower magnitude forces and impact self-selected speed gait less than the other muscles crossing the knee (Figure 6). Peak force for the BFSH was only about one-seventh of the SMEMB peak force (Figure 6). The BFSH may augment mid-swing knee flexion. Gracilis has been shown to contribute to hip joint flexion and adduction [15,55], with a negligible contribution to knee joint flexion [56]. Similarly, Sartorius generates only enough force to trivially enhance early swing knee flexion. While TFL crosses the knee, its force production is comparable to other hip abductors (GMED, GMIN), and has been discussed in depth elsewhere [57].

The popliteus has connective tissue attachments that border both the MCL and LCL and is ideally positioned—through the activation of muscle spindles and golgi tendon organs—to provide instantaneous feedback related to knee joint position [58,59]. Its primary function may not be related to wholly producing or restraining movement but rather to provide real-time knee position-related data. The finding that it is uniquely continuously active suggests a feedback role as its primary function [58]. POP was notable for nearly continuous low-amplitude EMG activity (Figure 7). A similar finding for popliteus activity in level walking is previously reported [60]. Its force output is certainly less than other muscles crossing the knee, and its role may be to prevent the forward dislocation of the femur on the tibia during the flexed knee stance. Assessment of the POP architecture suggests that the ratio of physiological cross-sectional area to muscle fiber length allows force production over a limited distance [61].

### 5.3. EMG and Muscle Force Variability in Selected Muscles

After accounting for electromechanical delay, the force produced by each muscle closely correlated with EMG activation. Most knee muscles showed remarkably uniform EMG timing across study participants, in other words, EMG onset, cessation and timing of peak relative intensity at similar percent gait cycle times for all tested individuals. However, there were a few muscles with slight variations in the EMG waveform across study participants.

The STEND onset times were predictable and consistent for all study participants. A few individuals had a slightly prolonged cessation, typically of low relative EMG intensity (and presumably low force output).

Swing phase activity in the short head of the biceps femoris was consistent across study participants. The non-hamstring BFSH stance phase activity was more variable, with slightly less than half of the tested individuals having stance phase EMG (and muscle force). Biceps femoris short head EMG profiles were averaged across all study participants. As a result of this EMG variability (and by augmentation, muscle force), the across-subject BFSH force estimates in stance may overestimate the force produced by study participants with only swing phase EMG activity and slightly underestimate the single support force produced by study participants with stance phase BFSH EMG activity. Stance phase BFSH EMG is not essential for normal gait kinematics, and its absence does not necessarily indicate gait pathology.

### 5.4. Potential “Error” Sources

One aim of this study was to determine if an EFP approach could yield reasonable estimates of individual knee muscle forces. The closeness of fit between the EFP and KIN power curves indicates that this model’s knee force estimates are acceptable. One potential advantage of the EFP approach is that the contribution of individual muscles to generation or restraint of limb movement can be determined. It is important to note that all input variables to the model (%MVC, EMD) were determined a *priori*—that this approach was not a curve-fitting exercise. This is an important distinction. The muscle model individual muscle force-time histories were determined, and activation patterns were directly measured (not predicted via imposed optimization criteria). Further, individual MTU kinematics were used as input to the model, and co-contraction of agonists and antagonists (readily observed with real-time gait EMG) was allowed when the joint moment was calculated. The a *priori* approach was applied to both the EMG input and the muscle model. Other models have used EMG activation to “drive” the model that may not match that seen in normal gait. Arnold [62] refined muscle activation patterns “as needed”. Their model was generally based on EMG timing that is observed in normal adults in gait. However, one controversial aspect of their model was the assumption that the vastus muscles did not produce substantial force(s) in swing. EMG recordings in normal adults clearly demonstrate knee extensor EMG in late swing, and even accounting for electromechanical delay knee extensor force production should be expected. While peak vastus forces occur in the weight-acceptance portion of stance, force output of more than 100 N was estimated in each of these knee extensors.

It is feasible that the match of actual versus predicted muscle forces is closer for some muscles than others. Muscle physiological cross-sectional area (PCSA), muscle fiber type, muscle length, tendon slack length, rate of change in MTU length and activation dynamics can influence muscle force. Of these, PCSA [63], optimal muscle fiber length [64], fiber type and pennation angle [63,65] and tendon slack length [29] are reasonably well established. Activation dynamics and the rate of change in MTU length are less well defined in gait and warrant further investigation. Accounting for EMD the timing of the EMG input to the force processor model closely matches the KIN power curves. Estimates of muscle force per percent gait cycle are dependent on obtaining a true maximum voluntary contraction (100% MVC) to scale the gait EMG.

The net EFP knee moment was the summed moment produced by the individual muscles. Each muscle’s moment contribution was the product of that muscle’s force output and moment arm. Both muscle force and moment arm magnitude could conceivably contribute to errors in the EFP model. Yet it is unlikely that there were substantial errors in moment arm magnitudes [57]. Hence, errors in individual muscle moment contributions are more likely related to over- or under-estimates of MTU forces. Force estimates were based on muscle fiber type, physiological cross-sectional area, total and tendon slack length and activation dynamics. Bogey [14,52] previously found that activation dynamics were the largest “error source” in a similar model that predicted gait cycle ankle forces and power.

Both the timing of muscle activation (EMG input, after accounting for EMD) and magnitude of the EMG signal affect force estimates. EMG relative magnitude has been shown to have the greatest effect on force estimates [15]. EMG relative magnitude during gait is based on that stride’s EMG intensity and the EMG magnitude obtained in the reference maximum muscle test. The ability to obtain a maximum voluntary contraction (MVC) can be technically challenging. It is difficult to obtain an authentic, maximum isometric contraction in a muscle that produces substantial force (e.g., vastus lateralis). As a result, the force estimates for some of the muscles examined may be slightly overstated. One consequence could be an overestimate of the EFP knee moments. Yet, EFP peak moments were of smaller amplitude than KIN moments, which suggests that the EFP force estimates were not adversely affected by the challenge of obtaining a true maximum voluntary contraction.

### 5.5. Contribution of Non-Muscle Tissues to the Net Joint Moment

Non-contractile tissues (joint capsule) are known to make a substantial contribution to the net moment [66] and power at the hip joint [67]. The hip approaches its boundaries for passive sagittal plane motion at the hip in normal gait. However, neither the ankle or knee come close to the bounds of passive motion in normal walking, and the potential contribution of knee passive elements is markedly reduced [68,69,70].

## 6. Conclusions

A consequence of good estimates of individual knee muscle force would be our enhanced understanding of that muscle’s role in efficient walking. Neuromuscular modeling techniques have produced reasonable estimates of ankle [14] and hip [15] muscle forces. The power produced at the adjacent joints during self-selected speed walking [2,52] has been described. Future work that determines the power produced by knee muscles would eliminate a substantial gap in the gait literature.

Knee force estimates are encouraging. However, the walking speeds examined are relatively slow. This model does not account for rapid change in MTU length, as this does not typically occur in self-selected speed walking. Extrapolation of the EFP approach presented here to more ballistic tasks (e.g., running) should be carried out with caution.

Knowledge of muscle forces crossing the knee is essential for other types of modeling of the joint. A future direction is to integrate the muscle forces presented here as part of a finite element model to estimate mechanical stresses across the tibial plateau during weight-bearing aspects of the gait cycle [71].

The presented model provides a reasonable estimation of in vivo knee muscle forces in self-selected speed walking. A desirable goal would be to expand this technique to the assessment of muscle forces in persons with gait pathologies. However, the underlying assumptions for the MTU anatomy and voluntary contraction estimates are not sufficiently individualizable. Further work, notably in scaling EFP models, is needed before any technique can be used to determine muscle forces (and etiology of abnormal gait) in persons with abnormal gait due to CNS pathology.

## Figures and Tables

**Figure 1 bioengineering-10-00980-f001:**
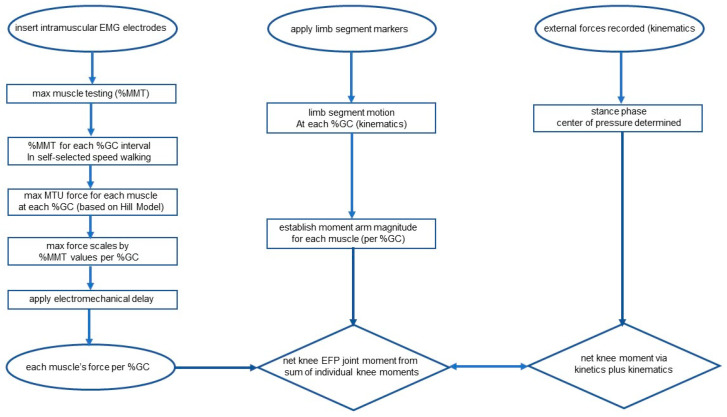
Flowchart depicting the process of the determining force for each tested lower extremity during gait (**left** column). Motion (**middle** column) and forces at each joint (**right** column) are also shown. Limb positions served as input for the Hill-based model, and peak forces were determined based on MTU length. Dynamic EMG was input to the model after accounting for electromechanical delay (EMD) and the moment produced by each muscle was the product of its force and moment arm at each %GC interval. The net EFP moment was the sum of the individual moments (middle column) and this was compared with knee moments obtained via inverse dynamics techniques (kinetics plus kinematics). Values were compared at 1%GC intervals.

**Figure 2 bioengineering-10-00980-f002:**
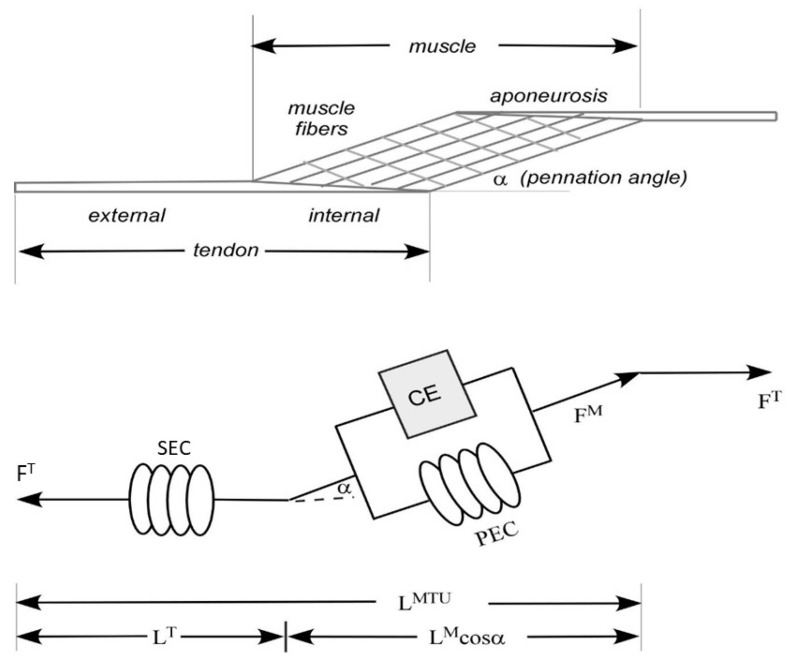
Hill’s muscle model is a three-element model consisting of a contractile element (CE) in series with an elastic spring element (SEC) and in parallel with an elastic parallel element (PEC). The model is a representation of the muscle’s mechanical response. Within this model, the estimated force–velocity relation for the CE element is modeled by Hill’s equation, which was based on tetanized muscle contractions where associated contraction velocities and loads were measured. CE contractile force is generated by actin and myosin links at the sarcomere level. The non-contractile elements (soft tissues) that surround the CE influence the muscle’s force–length curve. The PEC represents all passive forces of non-contractile tissues and has a mechanical behavior representative of soft tissues and is responsible for the muscle’s behavior when it is stretched. The SEC accounts for the inherent elasticity of the myofilaments.

**Figure 3 bioengineering-10-00980-f003:**
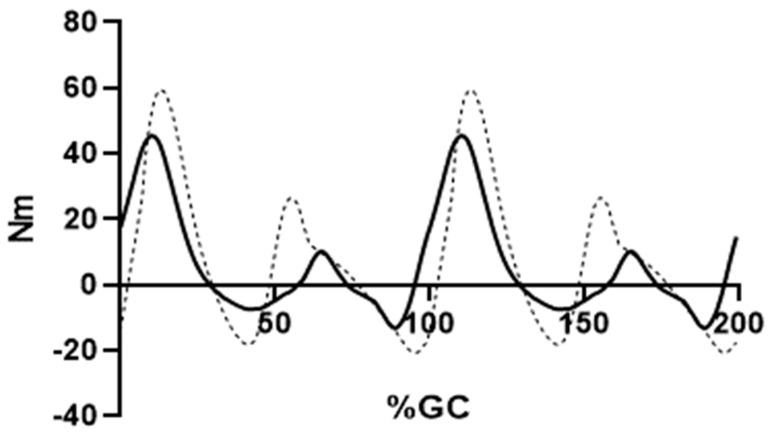
Knee moments for the EMG-to-force processing (EFP, solid line) and inverse dynamics methods (kinematics plus kinetics, (KIN, dashed line)) are shown. Two complete, consecutive strides are shown. Analysis period is from 63%GC (first stride) through 37%GC (second stride). Positive values indicate knee extension moments.

**Figure 4 bioengineering-10-00980-f004:**
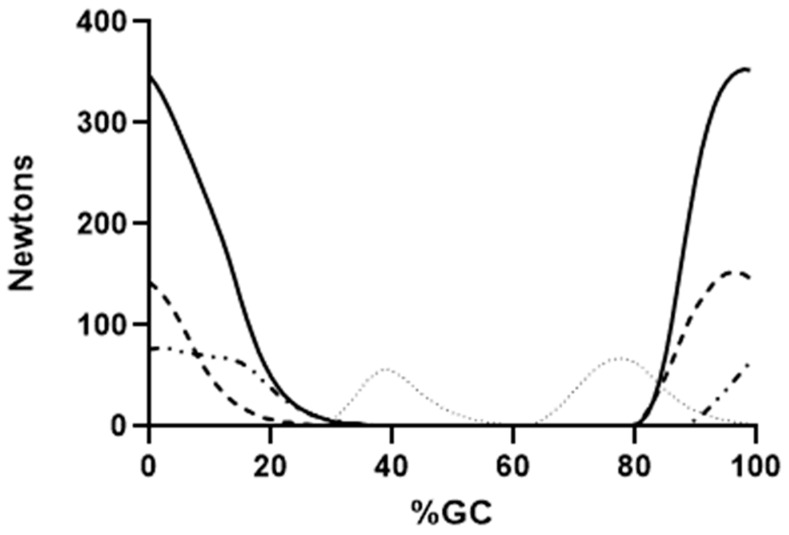
Mean EFP forces produced by the knee flexors are shown. Semimembranosis (solid line), semitendinosis (dashed line), long head of the biceps femoris (alternating dashed and dotted line) and short head of the bicep femoris (dotted line) are presented. Force production through the tendon is shown on the ordinate, percent gait cycle on the abscissa.

**Figure 5 bioengineering-10-00980-f005:**
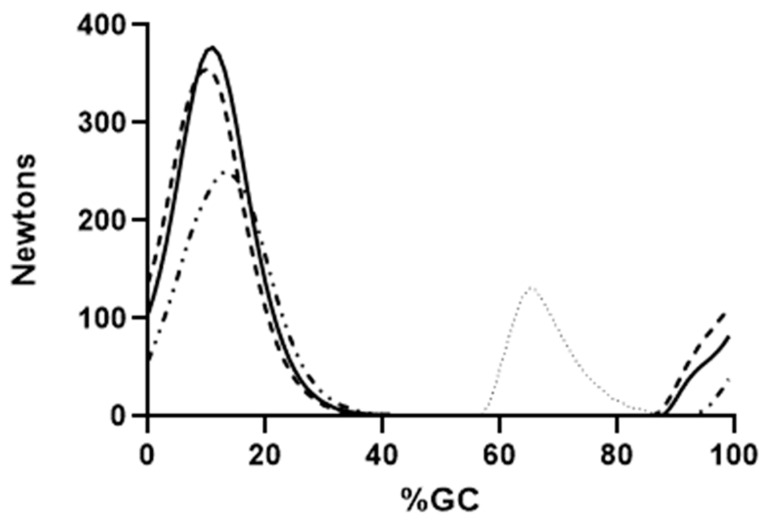
Mean EFP forces produced by the knee extensors are shown. Vastus lateralis (solid line), vastus lateralis oblique (dashed line), vastus intermedius (alternating dashed and dotted line) and rectus femoris (dotted line) are presented. Force production through the tendon is shown on the ordinate, percent gait cycle on the abscissa.

**Figure 6 bioengineering-10-00980-f006:**
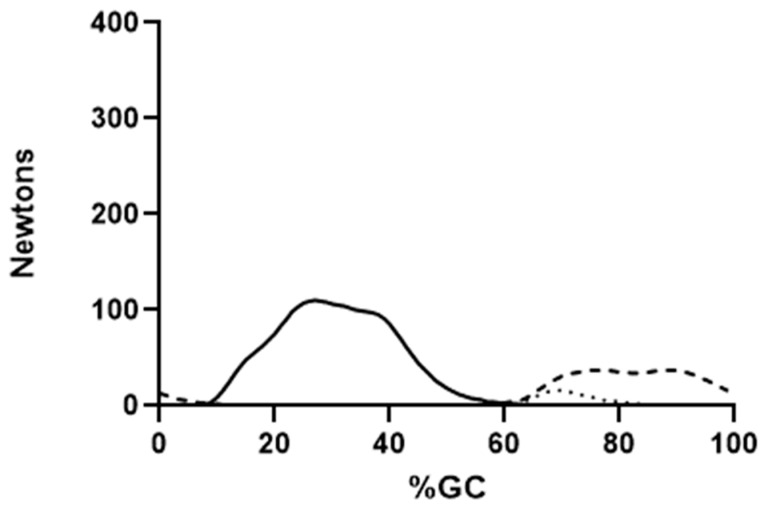
Mean EFP forces produced by muscles crossing the knee are shown. Tensor fascia lata (solid line), gracilis (dashed line) and sartorius (dotted line) are presented. Force production through the tendon is shown on the ordinate, percent gait cycle on the abscissa.

**Figure 7 bioengineering-10-00980-f007:**
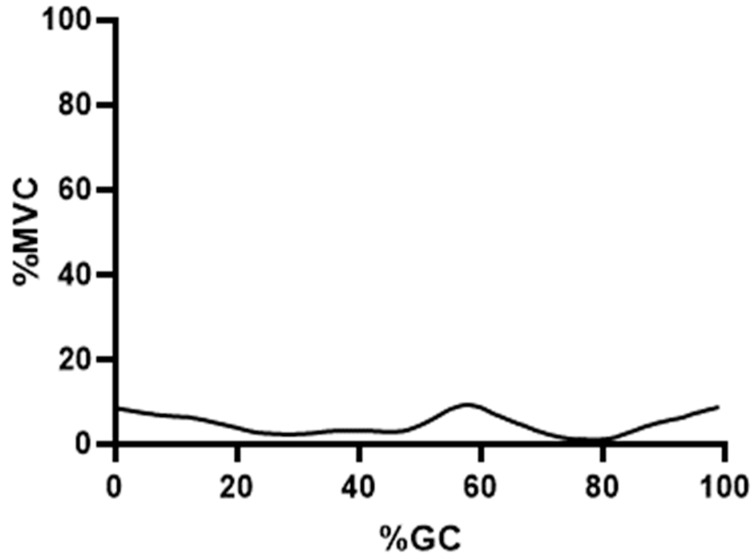
Mean percent maximum voluntary contraction is shown for the popliteus muscle (solid line). Relative EMG intensity is shown on the ordinate, percent gait cycle on the abscissa.

## Data Availability

Original data are available following reasonable request.

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
