# Peer review of "An EMG-to-Force Processing Approach to Estimating Knee Muscle Forces during Adult, Self-Selected Speed Gait"

_bioengineering, 2023, doi:10.3390/bioengineering10080980_

Round 1

Reviewer 1 Report

The EMG-to-force processing approach provides reasonable estimates of active individual knee muscle forces in self-selected speed walking in neurologically-intact adults. Some comments given as follows:

1.      Line 16, I think it should be revised from “hip position (angle of flexion (extension))” into “hip position flexion-extension”.

2.      Line 19, gold standard phrase is not scientific sound, recommended to change it.

3.      Line 27, What makes the author's novelty in the present article? My analysis suggests that other similar previous articles properly explain the points you have brought up in the current paper. Please be sure to emphasize anything truly novel in this work in the introductory section.

4.      Line 29, based on Jamari et al., gait is consisting of acted load, range of motion, cycle. Please provide this information along with relevant reference as follows: https://doi.org/10.1016/j.heliyon.2022.e12050

5.      Line 73, the using name of this subsection is not common, please review it as “Methods”.

6.      Line 97, the authors does not consistent for using “KIN” as initial, it is Kinematics and Kinetics as stated in line 97, kinetics plus kinematics as stated in line 18, or what? Please be consistent.

7.      Line 84, The basis for patient selection has to be more thoroughly explained by the authors. Has a protocol, foundation, or standard been adhered to? Given that the patients involved are heterogeneous and small, the current form was not unclear. It would have a direct effect on the outcomes that produce unsuitable work. One of the significant problems that the authors must resolve following the revision phase.

8.      Line 140, the explanation specific for this figure needs to provided in more comprehensive form.

-

Reviewer 2 Report

The proposed paper provides a unique and potentially informative technique for collecting and processing data for the estimation of knee muscle forces during gait.  The manuscript reflects a significant amount of well-informed research activity and describes in narrative form a commendable multi-step analysis process, reasonably well-documented. However, the lack of an overall view of the process may hinder an understanding of the flow of experiment and related analysis. To address this, the authors are encouraged to include a flowchart depicting the process, thereby facilitating a clearer understanding of the methodology.

There is no doubt that the Section 2 Methods is complete, and the manuscript weaves into the discussion a significant number of citations to contemporary reference papers, showcasing a strong foundation in existing research. However, the authors should be cautious about assuming too much prior knowledge from the reader. Clarifying concepts and providing context for complex references would improve the accessibility of the work and enhance the overall reading experience.   For example, Lines 76-79 are an important aspect of the research, but the terseness leaves the reader uncertain what aspects of analysis are repeated from the literature are represent unique aspects of the proposed analysis.  What is proposed as a new process for analysis is evidently the outcome of integrating prior work in new ways.  It is unclear if the intent of Lines 115-121 are new to the process or what one might typically use for center-of-pressure.  Lines 123-124 point to in-house software and is there anything unique to that which might hinder reproducibility of results.  It is not that specific steps are all that unusual or difficult to work out from the adequate and appropriate references provided, it is just difficult to integrate all of the information to achieve a full and clear understanding of the process. 

Section 2.4 presents a unique view of the classic Hill muscle model, though the description that follows has many, many steps  that are difficult to integrate – a more informative introduction would help.  When the reader gets to, say, Lines 180-183, the sentence refers to Reference 38 by Milner-Brown who has written extensive about the topic for decades, yet in the current work it is unclear if there is a “new twist” to an old theme or if this is used in a unique way, or even if this simply says it is a research standard. Later in that section, there are several steps that one would logically use in analysis, but the significance to the theme of the paper on a new process is unclear.

Overall, the challenge in reading the paper is that many traditional experimental and analytic steps are included in the research effort, but it remains difficult to walk away with a clear picture on what new and unique aspects are outcomes, or fi this is simply a use case for a newly developed process.  The paper speaks of developing a model, but what that model is does not emerge clearly and the reader grasps the value of the effort but a significant contribution to the field escapes visibility.  The authors are encouraged to reflect on the issues presented and suggested improvements to significantly strengthen the manuscript's contribution to the field.

Round 2

Reviewer 1 Report

Good effort in revision that have done to authors. Some following comments given in this form as follows:

1.      After reevaluate the manuscript, I have major concern about something new apart from authors published work, like in reference number [2], [14], [15], [37], and [52]. The justification is mandatory since become basis of rejecting this paper.

2.      Line 24, Rearrange keywords alphabetically.

3.      Line 109, the author has been mentioned related to swing and stance phase. Please provide the explanation that gait is consist of swing and stance phase. Give the explanation along with supporting relevant reference as follows: https://doi.org/10.3390/biomedicines11030951

4.      Line 130-131, regarding the centre of pressure, it is still ambiguous in the model developed by authors. Giving illustration regarding it would giving better understanding to the reader. Please address it.

-

Author Response

Evaluate the manuscript, I have major concern about something new apart from authors published work, like in reference number [2], [14], [15], [37], and [52]. The justification is mandatory since become basis of rejecting this paper.

Reference #37 (Computer algorithms to characterize individual subject gait during gait) address only how gait should be processed across multiple gait strides to generate a representative EMG profile for each patient/study participant. This method varied from the common ensemble averaging in that linear interpolation (described here) of the EMG from each stride was used to generate an average EMG profile that matched the mean onset and duration and representative EMG relative intensity across multiple gait strides for each muscle. That work did not make any attempts to determine muscle forces at the knee or any other joint.

Refences #2 (Estimates of individual muscle power production in normal gait) and #52 (Determination of ankle muscle power in normal gait using an EMG-to-force processing approach) are limited to estimating the power produced by muscles at the hip (reference #2) and ankle (reference #52). Knee joint forces are estimated in the novel work submitted here.

Neither Refences #14 (An EMG-to-force processing approach for determining ankle muscle forces during normal human gait) and #15 (An EMG-to-force processing approach for estimating in vivo hjp forces in normal adult walking) address muscle forces at the knee. A Hill-based model was developed to estimate in vivo forces for all lower extremity muscle during typical activities, yet that model requires specific input for each joint, and that input is unique to this research. That is, muscle-tendon unit (MTU) paths and resulting MTU lengths were specific to the knee in this research. Further, the moment arm magnitudes for each tested muscle were dependent on the aforementioned MTU paths. EMG profiles, MTU lengths and moment arm magnitudes – the three most key elements in this Model – are unique to this work.

 Line 24, Rearrange keywords alphabetically.

A new list of Abbreviations replaces the previous version. Please see new text in RED

Line 109, the author has been mentioned related to swing and stance phase. Please provide the explanation that gait is consist of swing and stance phase. Give the explanation along with supporting relevant reference as follows: https://doi.org/10.3390/biomedicines11030951

The presence of two distinct gait phases (stance and swing) is now integrated into the revised text. Please see new text in RED: “Normal gait consists of two phases: stance phase and swing phase. These phases can be further divided into a total of eight sub-phases [21]. The stance phase normally occupies about 60% of the total gait cycle [21] during which at least some part of the ipsilateral foot is in contact with the ground. The swing phase occupies the remaining 40% of the gait cycle [21].”

Line 130-131, regarding the centre of pressure, it is still ambiguous in the model developed by authors. Giving illustration regarding it would giving better understanding to the reader. Please address it.

I apologize for any confusion related to the center of pressure (CoP). The location of the center of pressure is NOT a part of the model. A revised Figure 1 has been inserted into the manuscript which correctly reflects that the CoP location affects only the net knee moment via kinematic and kinetics (used to validate the Model). Its precise location has no bearing on the knee muscle forces that are determined via EMG-to force processing.

Reviewer 2 Report

Revised manuscript addresses critical issues.

Author Response

REVIEWER #2

No changes suggested – revised manuscript addresses critical issues.
